Simultaneous bilateral laser therapy accelerates recovery after noise-induced hearing loss in a rat model

Lee Jae-Hun 1
Chang So-Young 1
Moy Wesley J. 2
Oh Connie 2
Kim Se-Hyung 3
Rhee Chung-Ku 4
Ahn Jin-Chul 5
Chung Phil-Sang 1 4
Jung Jae Yun 1 4
Lee Min Young eyeglass210@gmail.com 4
1 College of Medicine, Dankook University, Beckman Laser Institute Korea , Cheonan , South Korea
2 Beckman Laser Institute and Medical Clinic, University of California , Irvine , CA , United States
3 Department of Otolaryngology-Head and Neck Surgery, Jeju National University School of Medicine , Jeju , South Korea
4 Department of Otolaryngology-Head & Neck Surgery, College of Medicine, Dankook University , Cheonan , South Korea
5 Department of Biomedical Science, College of Medicine, Dankook University , Cheonan , South Korea
Fitzgerald Melinda
Electronic publication date: 2016 Jul 21
Publication date: 2016
Volume: 4
Electronic Location ID: e2252
Received 2016 May 10; Accepted 2016 Jun 23
Copyright: ©2016 Lee et al.
Copyright year: 2016
Copyright holder: Lee et al.
License: This is an open access article distributed under the terms of the Creative Commons Attribution License, which permits unrestricted use, distribution, reproduction and adaptation in any medium and for any purpose provided that it is properly attributed. For attribution, the original author(s), title, publication source (PeerJ) and either DOI or URL of the article must be cited.
License URL: https://creativecommons.org/licenses/by/4.0/

Keywords: Bilateral LLLT, Noise induced hearing loss, ABR, Hair cell survival

Funding: Ministry of Science, ICT and Future Planning grant NRF-2012K1A4A3053142 This study was supported by a grant of the Ministry of Science, ICT and Future Planning grant funded by the Korea government (NRF-2012K1A4A3053142) The funders had no role in study design, data collection and analysis, decision to publish, or preparation of the manuscript.

==============================
Noise-induced hearing loss is a common type of hearing loss. The effects of laser therapy have been investigated from various perspectives, including in wound healing, inflammation reduction, and nerve regeneration, as well as in hearing research. A promising feature of the laser is its capability to penetrate soft tissue; depending on the wavelength, laser energy can penetrate into the deepest part of the body without damaging non-target soft tissues. Based on this idea, we developed bilateral transtympanic laser therapy, which uses simultaneous laser irradiation in both ears, and evaluated the effects of bilateral laser therapy on cochlear damage caused by noise overexposure. Thus, the purpose of this research was to assess the benefits of simultaneous bilateral laser therapy compared with unilateral laser therapy and a control. Eighteen Sprague-Dawley rats were exposed to narrow-band noise at 115 dB SPL for 6 h. Multiple auditory brainstem responses were measured after each laser irradiation, and cochlear hair cells were counted after the 15th such irradiation. The penetration depth of the 808 nm laser was also measured after sacrifice. Approximately 5% of the laser energy reached the contralateral cochlea. Both bilateral and unilateral laser therapy decreased the hearing threshold after noise overstimulation in the rat model. The bilateral laser therapy group showed faster functional recovery at all tested frequencies compared with the unilateral laser therapy group. However, there was no difference in the endpoint ABR results or final hair cell survival, which was analyzed histologically.

Introduction

Hearing loss, caused by diverse factors, is an important public health issue. In particular, noise overexposure is considered harmful to hearing function. Intense noise can cause damage to hair cells by increasing oxidative stress, which produces various reactive oxygen species (ROS), such as the superoxide anion (O2−) (Yamane et al., 1995) and hydrogen peroxide (H2O2) (Ohinata et al., 2000).

Noise exposure can cause a temporary threshold shift (TTS) or a permanent threshold shift (PTS) that will not recover. The type of threshold shift is determined by the intensity and duration of exposure. Several studies with similar levels of noise and exposure times (>100 dB, >6 h) have reported that a PTS occurred after a few minutes or hours of such noise exposure (Buck, 1981; Hu et al., 2000; Hu, Henderson & Nicotera, 2006). Both TTS and PTS can occur simultaneously at different frequencies in one cochlea. According to recent research, damage to the auditory neurons, such as at the ribbon synapse and postsynaptic receptors, was found following noise exposure, even after recovery of the hearing threshold (Kujawa & Liberman, 2009).

Laser therapy has been used as a treatment for various symptoms, and its use has been increasing because of its non-invasive nature. After it was approved by the United States Food and Drug Administration, applications of laser therapy have widened in research scope, including wound healing (Anneroth et al., 1988; Grossman et al., 1998; Kana & Hutschenreiter, 1981), inflammation reduction (Boschi et al., 2008; Ferreira et al., 2005), and nerve regeneration (Miloro et al., 2002; Mohammed & Kaka, 2007). The effects of laser therapy have also been reported in the area of hearing research. Some studies have demonstrated significant effects in reducing tinnitus and increasing auditory neuron activation (Littlefield et al., 2010; Medalha et al., 2012; Park et al., 2013). Recently, our group reported a promising recovery effect of laser therapy on cochlear hair cells in an animal study (Rhee et al., 2012). Tamura et al. (2015) also reported a cytoprotective effect of laser therapy in cochlear hair cells against noise overstimulation (Tamura et al., 2015).

One useful feature of the laser is the capability to penetrate soft tissue; depending on the wavelength, laser energy can penetrate into deep parts of the body without damaging non-targeted soft tissues. This enables the delivery of laser energy from multiple points, which may lead to faster or increased effects of the laser in the target area. In our previous animal experiments, we found improvements in the hearing threshold not only in the laser-irradiated group but also in the contralateral ear (Rhee et al., 2012). This suggests that unilateral laser therapy may affect the contralateral auditory organs. Thus, we measured the degree of laser penetration in the contralateral ear of SD rats and assessed the benefit of simultaneous bilateral laser therapy compared with unilateral laser therapy versus a control group.

Materials and Methods

Animal subjects

Male Sprague Dawley (SD) rats (180–200 g) were used in this study. Eighteen rats were randomly divided into three different groups (noise only (n = 6), unilateral laser (n = 6), and bilateral laser (n = 6)). All animals were treated in accordance with the Guide for Care and Use of Laboratory Animals (7th edition, 1996), as formulated by the Institute of Laboratory Animal Resources of the Commission on Life Sciences. All procedures were approved by the Institutional Animal Care and Use Committee for Dankook University (DKU-15-048).

Acute acoustic trauma

The acoustic stimulus was a narrow band of noise which has frequency information centered at 16 kHz with 1 kHz of bandwidth (116 dB SPL). Rats were placed in individual cages to prevent defensive behaviors and these cages were placed in acryl reverberant chambers with a speaker BEYMA CP800Ti (Beyma, Valencia, Spain) attached on top. The traumatic stimulus was generated with a type 1027 sine random generator (Bruel and Kjaer, Denmark) and amplified with a R300 plus amplifier (Inter-M, Seoul, Korea) for 6 h. For real time monitoring, a frequency-specific sound level meter (Sound Level Meter—Type 2250; Bruel and Kjaer, Copenhagen, Denmark) was used to monitor noise level in the chamber (placed on the floor) every hour so that consistent intensity (116 dB SPL) was maintained during noise exposure.

Auditory brainstem response measurement

Auditory brainstem responses (ABR) were measured to identify degrees of hearing loss and recovery. The evoked response signal-processing system (System III; Tucker Davis Technologies, Alachua, Florida) was used for ABR measurement. Animals were anesthetized with Zolazepam (Zoletil, Virbac, Carros Cedex, France) and Xylazine (Rompun, Bayer, Leverkusen, Germany) and placed in a sound proof chamber. Three needle electrodes were inserted at vertex (active) and beneath of each pinna (reference and ground), subcutaneously. The tone-burst stimuli (4, 8, 12, 16, and 32 kHz) were used for experimental measurements and a total 1,024 responses were averaged. Responses were measured in 5 dB intervals from 90 to 10 dB SPL and thresholds were determined by the presence of peak within each signal. Hearing thresholds were obtained before and after noise exposure (Fig. 1). ABR measurement was also performed during and after laser irradiation (after the 3rd, 6th, 9th, 12th, and 15th laser irradiations).

Figure 1 Results of ABR measurement.

Consistent peaks were recorded at 16 kHz during ABR measurement as a baseline (A). After six hours of noise exposure, the overall amplitude of the peaks were reduced compared to the baseline result, and the peaks disappeared under 65 dB SPL at the same test frequency (B).

Laser irradiation treatment

An 808 nm diode laser (Wontec, Seoul, South Korea) was used for laser therapy. Each rat in the experimental group was anesthetized and irradiated for 60 mins (165 mW/cm2, 594 J) for 15 days. The density of the laser was calibrated with a laser power meter (FieldMax II-To, Coherent, USA) and detector sensor (Powermax; Coherant, Santa Clara, CA, USA). The optical fiber (core fiber 62.5 µm, cladding 125 µm) was attached to a hollow tube and placed into the external ear canal while leaving a distance of 1 mm between the fiber tip and tympanic membrane. Laser irradiation was performed on both the right and left ear simultaneously for the bilateral group and only in the right ear for the unilateral group. The noise only group was anesthetized and the optical fiber was placed into the external ear canal without power. Additional detailed information of the laser is described in Table 1.

Table 1 Laser (Photobiomodulation) parameter.

Parameter	Laser group (Bilateral and Unilateral)	
Power (mW)	185	
Beam spot size at target (cm2)	0.22	
Irradiance at target (mW/cm2) power density	841	
Exposure duration (s)	3,600	
Radiant exposure (J/cm2) fluence	2,700	
Radiant energy (J)	594	
Number of points irradiated	1	
Area irradiated (cm2)	0.22	
Application technique	Through tympanic membrane	
Number and frequency of treatment sessions	Once a day for 15 days	
Total radiant energy (J)	8,910	

Measurement of laser energy in the contralateral ear

Laser energy was first measured from the contralateral side of the ear using an 808 nm laser irradiation (calibrated as 165 mW) in the SD Rat to confirm the delivery of laser energy to the contralateral cochlea. The rat was sacrificed in a CO2 chamber and a secondary decapitation was performed to ensure the rat was no longer alive. The skin and pinna of the test ear (contralateral side from the laser irradiation) were removed to expose the cochlea. The exposed contralateral cochlea was placed above the laser detector and the laser was used to irradiate the ipsilateral external canal with the protocol explained above.

Hair cell count

For the quantitative analysis of outer hair cells (OHCs), whole mounts of the organ of Corti were prepared. Intracardiac perfusion was performed using 4% Paraformaldehyde (PFA) followed by 0.9% normal saline. The cochlea was then harvested. After harvesting, the cochlea was fixed in 4% PFA overnight. After washing with 0.1 M Phosphate-buffered saline (PBS), the cochlea was decalcified with ethylenediaminetetracetic acid (0.5 M EDTA, pH 8.0) and was dissected into three parts. The samples were prepared by staining with Phalloidin (Phalloidin-FITC, Sigma, St. Louis, MO, USA) and rinsed with 1x PBS. The samples were carefully examined under confocal microscopy (LSM 510 META, Zeiss, Germary) at a magnification of 400X.

We chose three representative areas for the quantitative analysis of OHC, which were located at 20, 50, and 80% from the apex, representing 4, 12, and 32 kHz respectively (Viberg & Canlon, 2004). Hair cells <200 µm in length were counted in each representative area. The morphometric analysis software Image J (http://rsb.info.nih.gov/ij/) was used to count the number of cells in each section.

Statistical analysis

All data were analyzed statistically using the Statistical Package for the Social Sciences software (SPSS, Version 19, IBM, Somers, USA). We performed a Tuckey post hoc test following a Two-way Analysis of Variance (ANOVA) to determine the significance between hearing threshold for ABR measurement and number of hair cells.

Results

Energy from the 808 nm laser was detected in the contralateral ear

The 808 nm laser energy was first measured in the contralateral ear. Using an open air setup between the laser probe and detector, the energy output was determined to be the same at the detector and the displayed output of the laser. A total of 6 mW of laser energy was measured in the detector at the contralateral ear, while the maximum level of laser energy penetrating the contralateral ear was found to be 8 mW. This result suggests that some laser energy was absorbed prior to exiting the other ear (contralateral ear).

Hearing loss after noise overstimulation

ABRs were measured before noise exposure to determine the baseline hearing threshold. Mean values (SDs) were 18.61 (5.37), 16.11 (5.57), 16.94 (6.67), 16.11 (5.3), and 16.39 (6.14) at frequencies of 4, 8, 12, 16, and 32 kHz, respectively (Fig. 2). At 24 h after noise exposure, ABRs were measured again to confirm the degree of hearing loss. Hearing thresholds were increased markedly after noise exposure. Mean values (SD) were 51.11 (6.08), 57.78 (8.44), 60.28 (6.96), 63.06 (4.79), and 60.56 (4.82) at frequencies of 4, 8, 12, 16, and 32 kHz, respectively (Fig. 2). Thus, these results indicate that overstimulation with a stimulus of 115 dB SPL can cause PTS.

Figure 2 Changes in hearing threshold at each ABR measurement.

At every tested frequency, the result of the bilateral LLLT group showed faster hearing recovery than the unilateral LLLT group, asterisk represents the statistical difference between the two groups noted above itself (NE, Noise Exposure; BC, Bilateral and Control; BU, Bilateral and Unilateral; UC, Unilateral and Control).

Laser therapy improved hearing recovery in the bilateral and unilateral treated groups

After the sixth laser irradiation, there was a significant difference in the hearing threshold at 16 and 32 kHz between the noise-only and the bilateral laser-treated groups (p = 0.001 at 16 kHz and 0.046 at 32 kHz; Figs. 2D and 2E). After the ninth laser irradiation, significant differences existed at all test frequencies between the noise-only and the bilateral laser-treated groups (p = 0.009 at 4 kHz, 0.04 at 8 kHz, <0.001 at 12 kHz, 0.001 at 16 kHz, and <0.001 at 32 kHz)(Fig. 2). The response of the unilateral laser-treated group was significantly different from that of the noise-only group at 32 kHz (Fig. 2E) after the ninth laser irradiation. The difference between the unilateral and the noise-only group increased to 12 kHz and 16 kHz after the twelfth laser irradiation, and the bilateral-treated group showed difference at all frequencies except 8 kHz (Figs. 2C and 2D). Finally, after the 15th laser irradiation, the hearing threshold at all test frequencies was significant different in the noise-only compared with the bilateral laser-treated group (p < 0.001 at 4 kHz, 0.005 at 8 kHz, <0.001 at 12 kHz, <0.001 at 16 kHz, and <0.001 at 32 kHz), and the difference between the unilateral group and noise-only group increased to 4 kHz (Fig. 2). This result showed that both bilateral and unilateral laser therapy could reduce the hearing threshold in the SD rat model after noise overstimulation. However, complete recovery of the hearing threshold (to the baseline level) was not achieved.

Bilateral laser therapy resulted in faster hearing threshold recovery than did unilateral laser therapy

A significant difference in the threshold between the bilateral group and the noise-only group was observed from the point of the sixth laser irradiation (at 16 kHz and 32 kHz) (Figs. 2D and 2E). In contrast, significant differences between the unilateral group and the noise-only group were observed from the points of the ninth and twelfth laser irradiations (at 32 kHz and 16 kHz; Figs. 2D and 2E). Furthermore, compared with the hearing threshold recovery in the bilateral group at 4 kHz, 8 kHz, and 12 kHz after the ninth laser irradiation, hearing threshold recovery in the unilateral group at these frequencies (at 4 kHz and 12 kHz) was observed after the twelfth and 15th laser irradiations (Figs. 2A and 2C), respectively. At 8 kHz, there was no significant difference between the unilateral group and the noise-only group at any time point. This result indicated that despite the absence of differences in the extent of hearing recovery between the unilateral and bilateral laser therapy groups, the bilateral simultaneous application of laser therapy induced faster (up to 3 days) recovery of the hearing threshold after noise-induced hearing loss compared to the unilateral laser therapy group.

Laser-treated group showed better outer hair cell (OHC) preservation in the basal turn

A confocal image of three representative areas is presented in Fig. 3. At the apex and the middle area, the averages of OHCs were similar across the three experiment groups (73.67, 72, and 70.33 at the apex, and 71, 72.67, and 73 at the middle, in the bilateral, unilateral, and noise-only groups, respectively; Fig. 4). However, averages of OHCs at the basal turn were found to be different among each group (72.67, 67.5, and 59 in the bilateral, unilateral, and noise-only groups, respectively), and both the bilateral and unilateral laser groups showed larger number of OHCs than did the noise-only group (p = 0.0052 and 0.0006, respectively; Fig. 4).

Figure 3 Representative confocal images of hair cells at three different locations (apex, middle, and base) in each experimental group.

Missing hair cells were observed only at the base part of the cochlea in the noise-only group.

Figure 4 Numbers of OHCs in three parts of the basilar membrane in each group.

The bilateral and unilateral laser groups showed significantly larger numbers of OHCs at the base part of the basilar membrane (**p < 0.01, ***p < 0.001).

Discussion

Cochlear damage can be variable, and a hearing threshold shift can occur abruptly or progressively, depending on the intensity and duration of noise overstimulation (Clark, 1991). In the results of the present study, we found permanent threshold shifts in almost every frequency region examined. These results are consistent with our previous study (Rhee et al., 2012). The results demonstrate that a high level of noise can cause PTS in this rat model. We observed slight improvements in the hearing threshold at low-frequency regions (4 and 8 kHz) with no treatment, which could be explained as a TTS, because it was not the main target frequency (Clark, 1991) of the acoustic overstimulation applied in the current study. Increases in hearing threshold after noise exposure as both PTS and TTS could be a result of loss or dysfunction of OHC electromotility, which contributes to hearing sensitivity by amplifying the incoming stimulus (Liberman et al., 2002). However, in the present study, we found that the loss of hearing function was not obviously correlated with the histopathology of the OHCs. For this functional loss, we hypothesize that there is some other mechanism of TTS or PTS that may be involved, such as the dispersal of presynaptic ribbons and postsynaptic receptors, which connect the inner hair cells and spiral ganglion (Furman, Kujawa & Liberman, 2013).

Application of laser therapy, after noise overstimulation, induced recovery of hearing function similar to our previous study (Rhee et al., 2012). This protection mechanism is considered to be related to the inhibition of iNOS and caspase 3 expression (Tamura et al., 2015), but the details of the underlying mechanism remain unclear. Also, another theory is that this effect may be explained by the balance of free radicals and antioxidants. Before hair cell death, ROS levels increase as a result of noise overexposure. Movement of electrons in hair cells releases energy for converting adenosine diphosphate (ADP) to adenosine triphosphate (ATP) by phosphorylation. During this process, superoxide is generated as an intermediary. When the use of oxygen is increased due to noise exposure, the generation rate of superoxide is also increased by the activity of the mitochondria (Evans & Halliwell, 1999). During noise exposure, mitochondria are strongly stimulated and produce excessive superoxide as a byproduct. Superoxide can react with other molecules in cochlear hair cells, resulting in subcellular molecular damage. Decreased cochlear blood flow due to noise exposure can also contribute to a deficiency of oxygen in the cochlea. Increased ROS can damage DNA, lipids, and proteins, leading to hair cell death (Evans & Halliwell, 1999).

Despite the low penetration level in the contralateral ear, we found faster hearing recovery in the bilateral laser therapy group compared to the unilateral laser therapy group. Additional laser energy may improve the speed of hearing recovery by prompting the endo-organs of the contralateral cochlea. We hypothesize that the laser energy penetrating directly can affect the contralateral cochlea as an activator of cell metabolism. Additionally, the amount of laser energy in the middle of the head, which would be more than that reaching the contralateral cochlea, could have sufficient influence to activate the cochlear nerve or auditory pathway in the midbrain. Moreover, protective actions of cochlear efferent feedback pathway through olivocochlear bundle may also increase recovery. Several studies suggest that the protective effect of the olivocochlear bundle during hearing is allowed to suppress hyper activations of auditory nerve fibers and IHCs (Gifford & Guinan, 1987; Guinan & Gifford, 1988). A damaged olivocochlear bundle by surgical lesion can cause alteration of ABR response, resulting in the increase of hair cell vulnerability related to noise exposure (Maison, Usubuchi & Liberman, 2013). Depositing more laser energy into the olivocochelar bundle in the bilateral laser group may increase this protective effect, resulting in a faster recovery compared to unilateral laser group. Additionally, the bilateral laser therapy group did not show improved recovery of the hearing threshold compared to the unilateral laser therapy group after the 15th laser irradiation. This limited effect can be explained by the destruction of the most vulnerable auditory pathways after noise exposure, such as synaptic ribbons (Kujawa & Liberman, 2009). Relatively normal morphologies of OHCs after noise overexposure support this hidden damage theory because the functional loss was much more dramatic compared to the apparently limited hair cell loss found in the histology. There may be additional mechanisms responsible for the functional loss of hearing after noise overexposure, such as synaptic degeneration (Kujawa & Liberman, 2009).

The faster effect of bilateral laser therapy versus unilateral laser therapy is promising for clinical use. Most treatments after hearing loss due to different insults, require early intervention (Ward, 1960). There are critical periods of time that can increase the success of a treatment outcome, resulting in a more favorable prognoses (Chen et al., 2007). With bilateral laser therapy, a shorter time frame was required to achieve a desirable outcome; thus, there is a higher chance of staying within the “golden time” for the treatment of hearing loss. Transcanal laser therapy treatments can lead to middle ear complications, such as acute inflammation and perforation of the tympanic membrane (Moon et al., 2016). Applying bilateral laser therapy may reduce the possibility of complications while increasing the effect because laser energy is delivered from two different sites, similar to the protocol for transcranial laser therapy. Multiple site laser irradiation has been used for transcranial laser therapy by several groups (Barrett & Gonzalez-Lima, 2013; Schiffer et al., 2009). These studies reported improvements in cognitive and emotional functions in the brain, with no side effects due to laser irradiation, using lower laser power and irradiating from multiple positions. As such, if estimating the exact location of the cochlea is possible, we may be able to deliver energy to the cochlea from multiple sites transcranially. However, no methodology for transcranial aiming toward the cochlea has yet to be established or developed.

To apply bilateral laser therapy in the clinic, some practical issues must be considered. Because of anatomical differences between humans and rodents, the effects of laser energy on the contralateral side would be different in humans compared to rodents. The larger distance from one ear to the other may limit the delivery of laser energy; however, the beneficial effect of bilateral laser therapy would be expected to remain if the mechanism involves targeting the brainstem. Increasing the power of the laser may be another approach to deliver energy to the other ear, but this could cause unwanted side effects, resulting in local burning and tympanic perforation. Thus, increasing the power of transcanal laser irradiation should be carefully considered before translation to clinical application.

Conclusions

The present study showed positive effects of bilateral laser therapy after noise-induced hearing loss in an animal model. The results suggest that the use of bilateral laser therapy in a clinical setting may improve the therapeutic effects on hearing while minimizing side effects.

Supplemental Information

Data S1 ABR thresholds

Click here for additional data file.

Data S2 individual ABR thresholds

Click here for additional data file.

Additional Information and Declarations

Competing Interests

Author Contributions

Animal Ethics

Data Availability

The authors declare there are no competing interests.

Jae-Hun Lee conceived and designed the experiments, performed the experiments, wrote the paper, prepared figures and/or tables.

So-Young Chang performed the experiments, analyzed the data, contributed reagents/materials/analysis tools.

Wesley J. Moy and Min Young Lee wrote the paper, reviewed drafts of the paper.

Connie Oh reviewed drafts of the paper.

Se-Hyung Kim, Jin-Chul Ahn and Phil-Sang Chung analyzed the data.

Chung-Ku Rhee and Jae Yun Jung conceived and designed the experiments.

The following information was supplied relating to ethical approvals (i.e., approving body and any reference numbers):

All animals were treated in accordance with the Guide for Care and Use of Laboratory Animals (7th edition, 1996), as formulated by the Institute of Laboratory Animal Resources of the Commission on Life Sciences. All procedures were approved by the Institutional Animal Care and Use Committee for the Dankook University (DKU-15-048).

The following information was supplied regarding data availability:

The raw data has been supplied as Supplemental Dataset.

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
