# Peer review of "Simultaneous bilateral laser therapy accelerates recovery after noise-induced hearing loss in a rat model"

_PeerJ, doi:10.7717/peerj.2252_

## Round 0.1 · original submission · Minor Revisions

· Academic Editor

Minor Revisions

It will be important to address the English language criticisms raised by several of the reviewers and also the potential double presentation of the results in Figures 2 and 3. Other comments raised by the reviewers could be addressed by additions to the Discussion, including further references. As a further comment, the current discussion of potential mechanism via actions on oxidative stress would be enhanced with more comprehensive referencing and relating to the findings of the current study.

Reviewer 1 ·

Basic reporting

The manuscript is fairly well written, but English sentence construction still need improvement. The main error grammatically is the very frequent omission of the required definite and indefinite articles (the , an, a) before nouns. I cannot believe that a literate native English would not have commented on this in early drafts and it needs to be corrected throughout the manuscript.

1. Should the units nm be included each time 808 later is mentioned?
2. Details of the loudspeaker used to deliver the noise trauma should be supplied
3. The Rhee et al references 2012a and b are identical. One should be removed, or the correct reference inserted
4. I do not think Figure 1 adds anything to the paper. It should be deleted provided the necessary details are included in the text.
5. It seems to me that Figures 2and 3 replicate the same data. Figure 3 is much easier to follow and unless the authors can explain what extra information it provides, Figure 2 could be omitted.
7. Figure 5, the vertical axis should state the units i.e. number of hair cell/200um?

Experimental design

no comments

Validity of the findings

It would be desirable to show some examples of the ABR waveforms so that the reliability of evaluating thresholds from Wave I could be demonstrated.

Comments for the author

This work follows on from previously published reports of a protective effect of 808nm laser on hearing function after noise exposure. Previous work involved unilateral irradiation in rats and this report uses the same basic procedure to compare the effects of unilateral and bilateral irradiation. Both irradiated groups showed faster recovery of auditory nerve thresholds than non-irradiated animals exposed to the same acoustic trauma (thresholds assessed using wave I of the ABR). It is reported that bilateral exposure results in faster recovery than unilateral exposure. The authors speculate that this may be due to summation of the beneficial effects caused by spread of the laser light across the head.
The Discussion should mention another possible mechanism, which is the activation of the well-known protective action of the olivocochlear efferent pathways (numerous papers by R Rajan and by the Liberman group). This would fit with the greater effect of bilateral irradiation since these pathways have both crossed and uncrossed components to each cochlea.

·

Basic reporting

No comments

Experimental design

No comments

Validity of the findings

No comments

Comments for the author

594 J/cm2 is a relatively high dose of 808 nm light relative to other studies using NIR light for treatment in animal models of Parkinson's Disease (4-10 J/cm2), Multiple Sclerosis (4 J/cm2) and retinitis pigmentosis (4.5 J/cm2). Have you studied the efficacy of lower doses. Perhaps a dose-response study is warrented.

Reviewer 3 ·

Basic reporting

This is a sensibly designed study, which tests whether low level laser irradiation will accelerate the recovery of auditory function, after thresholds have been raised by damaging noise.

Experimental design

The design is simple and sensible

Validity of the findings

I find the data clear, well presented and compelling

Comments for the author

The authors have done a good job with this study. It will build the scientific basis for using LLLT or photobiomodulation to reverse hearing loss. I would be most interested to learn in preconditioning is also protective - it is in many other models of red-light and tissue damage.
The authors have also done a good job with the language. I noted that I not find where they have defined ABR (that must me done). And the language could be improved by a sympathetic English-language editor. For example 'causes fast recovery' (in the title) would be better written as 'accelerates' recovery.

But the manuscript is clear nevertheless

---

## Round 0.2 · Minor Revisions

· Academic Editor

Minor Revisions

Thank you for your comprehensive revision; only one item remains outstanding. If it is considered essential to present the data of Figures 2 and 3 in two separate Figures, it is necessary to denote the additional statistical significances you refer to in Figure 3, using asterisks or similar on the graph. Without this, it is not necessary to present both Figures.

---

## Round 0.3 · accepted · Accept

· Academic Editor

Accept

The article has been appropriately revised and is now acceptable for publication.